# Iodine status of non-pregnant women and availability of food vehicles for fortification with iodine in a remote community in Gulf province, Papua New Guinea

Janny M. Goris[1], Victor J. Temple[2]*, Joan Sumbis[2‡], Nienke Zomerdijk[3‡], Karen Codling[4‡]

1 PNG Foundation, North Tamborine, Qld, Australia, 2 School of Medicine and Health Sciences, University of Papua New Guinea, Port Moresby, Papua New Guinea, 3 School of Medicine, The University of Queensland, Brisbane, Qld, Australia, 4 Iodine Global Network—Southeast Asia and the Pacific, Bangkok, Thailand

☯ These authors contributed equally to this work.
‡ These authors also contributed equally to this work.
* templevj@upng.ac.pg

**Data Availability Statement:** The data underlying this study contain potentially identifying participant information and are thus available only on request.

## Abstract

Adequate iodine status of women of childbearing age is essential for optimal growth and development of their offspring. The objectives of the current study were to assess the iodine status of non-pregnant women, availability and use of commercial salt, extent to which it is iodised, and availability of other industrially processed foods suitable for fortification with iodine. This prospective cross-sectional study was carried out in 2018 in a remote area in Gulf province, Papua New Guinea. Multistage cluster sampling was used to randomly select 300 women visiting local markets. Of these, 284 met study criteria of being non-pregnant and non-lactating. Single urine samples were collected from each of them. Discretionary salt intake was assessed; salt samples were collected from a sub-sample of randomly selected households. A semi-structured, pre-tested questionnaire to assess use and availability of commercial salt and other processed foods was modified and used. Salt was available on the interview day in 51.6% of households. Mean iodine content in household salt samples was 37.8 ± 11.8 ppm. Iodine content was below 30.0 ppm in 13.1% and below 15.0 ppm in 3.3% of salt samples. Mean iodine content of salt available at markets was 39.6 ± 0.52 ppm. Mean discretionary intake of salt per capita per day was 3.9 ± 1.21 g. Median UIC was 34.0 µg/L (95% CI, 30.0–38.0 ppm), indicating moderate iodine deficiency. For women with salt in the household, median UIC was 39.5 µg/L (95% CI, 32.0–47.0 µg/L), compared to median UIC of 29.0 µg/L (95% CI, 28.0–32.0 µg/L) for those without salt. This community has low consumption of iodised salt, likely due to limited access. Investigation of other industrially processed foods indicated salt is the most widely consumed processed food in this remote community, although 39.8% of households did use salty flavourings.

Requests for data may be sent to the PNG
Foundation at the University of Papua New Guinea,
School of Medicine and Health Sciences
(contact@pngfoundation.org.au). The authors
confirm data would be made available to
researchers interested in replication or verification
of the present study, or otherwise addressing a
legitimate research question.

**Funding:** The authors received no specific funding
for this work.

**Competing interests:** The authors have declared
that no competing interests exist.

## Introduction

During pregnancy and lactation, adequate intake and bioavailability of iodine are required for the biosynthesis of thyroid hormones, which are important for the regulation of growth and the healthy development of the nervous system of the foetus and infant, control of metabolic activities, developmental processes, and functions of the central nervous system [1–3]. The iodine status of non-pregnant women of childbearing age is important because it is the status of women entering pregnancy, when adequate maternal iodine nutrition is important for foetal development [1].

Iodine deficiency in women of childbearing age can cause infertility and also set the stage for miscarriage or stillbirth during pregnancy [1]. According to recent studies [4–6] mild iodine deficiency in pregnant women has been associated with lower child IQ, poor neuro-cognitive outcomes, inadequate language development, and symptoms of attention-deficit hyperactivity disorder. To avoid some of these negative impacts, maternal iodine deficiency should be corrected pre-conception to ensure availability of adequate amount of thyroid hormones during early gestation [1, 2, 4].

Salt iodisation, a policy of iodising all salt for human and animal consumption, is the recommended first-line strategy for the control and elimination of iodine deficiency among vulnerable groups in affected communities [1, 2]. The recommended WHO/UNICEF/IGN indicator for salt iodisation with regard to sustainable elimination of IDD is that > 90% of households use adequately iodised salt (i.e. with iodine content 15–40 ppm) [1]. Salt iodisation has been implemented in Papua New Guinea (PNG) since June 1995, following promulgation of the PNG Salt Legislation, banning the importation and sale of non-iodised salt [7]. It was incorporated into the PNG Food Sanitation Regulation issued in 2007 [8]. Systematic monitoring is required for effective implementation of the salt iodisation policy [1]. PNG implemented a National Nutrition Survey in 2005 in order to assess impact of the policy. In addition, several small studies in different locations have been undertaken.

Findings of the National Nutrition Survey in PNG in 2005 (PNG NNS, 2005) indicated that 92.5% of salt samples taken from households were adequately iodised. It further stated that iodine status was "adequate" among non-pregnant women of child-bearing age, with median Urinary Iodine Concentration (UIC) of 170 µg/L [9]. However, on the day of data collection, 38% of households had no salt in the household, and women in these households had lower iodine status than those in households with salt (median UIC of 114 µg/L and 203 µg/L respectively) [9].

Two mini-surveys on iodine status carried out after the PNG NNS found adequate iodine status of non-pregnant women in an urban area, the PNG National Capital District Port Moresby; median UIC of 163.0 µg/L in non-pregnant women in 2006 study and median UIC of 124.5 µg/L in lactating women in 2009 study [10, 11]. However, moderate iodine deficiency was found in non-pregnant women and school age children in another mini survey undertaken in 2015 in a remote and mountainous rural area of Kotidanga Rural Local Level Government (LLG), Kerema district, Gulf province, with median UIC of 32.0 µg/L and 36.0 µg/L in school age children and in non-pregnant women, respectively [12]. A larger and more representative follow-up survey, conducted in 2017 in the same community, found school age children to also be mildly to moderately iodine deficient (median UIC of 25.5 µg/L) [13]. Furthermore the two Gulf province studies found that the community has limited access to commercial salt, some of which was not adequately iodised, and that there was limited knowledge of the importance of using iodised salt and the consequences of iodine deficiency with regard to health outcomes [12] [13]

Considering currently available PNG data on iodine status and availability of iodised salt, it appears that, while the majority of household salt in the country is adequately iodised and iodine status of the general population is adequate, there are remote communities that have very limited access to commercial and, therefore, iodised salt, and that these remote communities, as a consequence, suffer from mild to moderate iodine deficiency. There is a need to learn more about the availability of commercial salt, its impact upon iodine status of populations and about alternative strategies to increase iodine intake in remote communities.

The current study therefore returned to the same remote community in Kotidanga Rural LLG, Kerema district, Gulf province to re-assess the iodine status of non-pregnant women of reproductive age, the availability of commercial salt and the extent to which it was iodised, and the availability of other industrially processed foods that might be fortified with iodine.

## Methods

### Study site and population

The study was carried out in Kotidanga Rural LLG, Kerema district, Gulf province, PNG, the same community of the two earlier studies [12] [13], which assessed iodine status in non-pregnant women and school age children in 2015 and 2017, respectively. Gulf is one of the 22 provinces in PNG; it shares land borders with six other provinces [14]. It has rugged mountainous landscapes, grassland flood plains and lowland river deltas. Gulf province has two districts, Kerema and Kikori. The capital of Gulf province is Kerema. The local population is mainly the "Kamea", which are members of the Angan-speaking tribal group [14]. The population of Kotidanga Rural LLG is 45,385 [15]. In Kotidanga Rural LLG, there is one hospital at Kanabea village, established by the Catholic mission in 1964; one Health Extension officer is servicing a population of around 20,000 Kamea people [15, 16]. Kotidanga Rural LLG is between 1200–1600 meters above sea level with temperatures between 12˚C and 30˚C and a yearly rainfall of 4000–7000 mm [17, 18]. The actual study site is a remote mountainous area with no road access; the only way of getting there is by walking along mountain paths or by air transport. It usually takes about 3 days walking during daylight from the location of the hospital to the closest settlement in Kerema, Gulf province or Menyamya, Morobe province.

### Sample size

According to the recently released UNICEF *Guidance on the Monitoring of Salt Iodization Programmes and Determination of Population Iodine Status* [19], "around 400 urine samples per population group are required to measure the median UIC with 5% precision, and 100 urine samples to measure the median UIC with 10% precision". In the current mini-survey with limited resources, a sample size of 300 non-pregnant women of childbearing age was considered adequate to provide sufficient precision to determine the median UIC.

### Study design and sampling

This was a prospective community based cross-sectional study carried out in Kotidanga Rural LLG conducted during March-April 2018. Five major markets in the community were selected to participate in the study. Multistage cluster sampling was used to randomly select 300 women, in the age group 15 to 45 years, visiting the major markets. Of the 300 women selected, 284 (94.7%) were non-pregnant, non-lactating women and 16 (5.3%) were pregnant women in their first-trimester. The 16 pregnant women were excluded from the study.

## Collection of salt samples

The objectives of the study were explained to the village authorities, who then communicated the information to the women and families in their communities. On the day of the interview in the market place, each of the selected women was asked if they have salt at home. The response of 149 (52.5%) of the 284 women was positive. Because of logistical reasons, it was not possible to visit all their households for collection of a salt sample; therefore, the study selected a sub-sample of 62 households (41.6%) from the 149 that reported they had salt in the home on the day of data collection. These 62 households were visited for collection of a teaspoon of salt.

One brand of commercial salt, sold only in clear high-density polyethylene packages of 250 g and 100 g, was available in all of the major markets in the study area. Packets of the salt were purchased for analysis from each of the following markets: Lot 1: Kotidanga (GPS coordinates: -7.386722, 146.003611), Lot 2: Kanabea (GPS coordinates: -7.5367451, 145.9051325) and Lot 3: Ipayu (GPS coordinates: -7.318624, 145.944936). Additional 100g packets of the same brand were purchased in the National Capital District (NCD) for comparison of iodine content. Subsequently, all the salt samples were analysed for their iodine content.

## Discretionary intake of salt

A subset of 20 households was randomly selected among the 62 households. A sealed 100 g packet of the same brand of salt found in the market was given to each of the 20 randomly selected households to use for food preparation and consumption as usual. The number of adults living in each household and eating food from the same cooking pot/hearth was counted and recorded. Each household was visited three days later to determine the amount of salt remaining in the packet. The number of adults living in each household was again counted and recorded. The data obtained was used to estimate the average discretionary intake of salt per capita per day.

## Urinary iodine concentration (UIC)

For the determination of UIC, single urine samples were collected at the markets from each of the 284 consenting non-pregnant women. Permission to use a safe and secured private location for collection of urine samples was approved by the authorities in each of the markets. Each urine sample was kept in a properly labelled sterile plastic tube with a tight-fitting stopper that was further sealed with special plastic band.

## Questionnaire on use of salty flavourings, commercial salt and other food vehicles for fortification with iodine

A semi-structured questionnaire used in earlier studies [20] was modified and adapted for use in this study. It was pre-tested in Kotidanga Rural LLG among 20 women 15 to 45 years old, recruited from five different villages in the district. Feedback and suggested changes were provided orally by the women and in English writing by interpreters. The feedback was used to adapt and improve the questionnaire. It was then used to assess use of salty flavourings, commercial salt and availability of other industrially processed foods.

The salt samples, urine samples and completed questionnaires were transported by airfreight to the Micronutrient Research Laboratory (MRL) in the School of Medicine and Health Sciences (SMHS) University of Papua New Guinea (UPNG) for analysis.

## Analysis of salt and urine samples

Quantitative assay, in duplicate, of iodine content in salt collected from the households and purchased in the markets was carried out, using the WYD Iodine Checker [21]. The Westgard Rules using Levy-Jennings Charts were used for internal bench quality control (QC) for daily routine monitoring of performance characteristics of the WYD Iodine Checker. The percent coefficient of variation (CV) ranged from 2.5% to 5.0% throughout the analysis.

The Sandell-Kolthoff reaction was used to determine the UIC in urine samples after digesting the urine with ammonium persulfate in a water–bath at 100°C [1]. The Levy-Jennings Charts and the Westgard Rules were used for internal bench QC characterization of the assay method. The sensitivity (10.0–12.5 μg/L) and percentage recovery (95.0 ± 10.0%) of the urinary iodine (UI) assay were frequently used to assess the performance characteristics of the assay method. External QC monitoring of the assay procedure was by Ensuring the Quality of Urinary Iodine Procedures (EQUIP), which is the External Quality Assurance Program (QAP) of the Centres for Disease Control and Prevention (CDC), Atlanta, Georgia, USA.

## Data analysis and interpretation

Statistical analyses of the data were carried out using the Statistical Package for Social Sciences (SPSS) software (version 17) and the Microsoft Excel Data Pack 2010. Normality of the data was assessed by the Shapiro-Wilks test. Mann Whitney U and Wilcoxon W tests were used for differences between two groups; Kruskal-Wallis, Friedman and bootstrapping were used as appropriate. A p-value of < 0.05 was considered as statistically significant.

The criteria used for interpretation of the salt iodine data were based on the PNG salt legislation [7, 8]. According to the legislation, all salt must be iodised with potassium iodate; the amount of iodine in "table salt" should be 40.0 to 70.0 ppm (mg/kg); the amount of iodine in "other salt" should be 30.0 to 50.0 ppm. "Other salt" in the legislation refers to non-table, edible salt e.g. crystal or coarse salt. These amounts of iodine should be present at production or import level. WHO recommendations for iodine levels of food grade salt aim to provide 150 μg iodine per day, assume 92% bioavailability, 30% losses from production to household level before consumption and variability of ±10% during iodisation procedures [22]. If 30% of iodine is lost from salt iodised as per PNG food regulations, iodine content of table salt at household level should be between 28 ppm (40 ppm minus 30%) and 49.0 ppm (70.0 ppm minus 30%). This implies that, in PNG, the iodine content in salt in retail outlets or at the time of consumption should be between 28.0 ppm and 49.0 ppm [7, 8]. A cut-off of 30.0 to 50.0 ppm has been used in the analysis of this study by rounding up the figures. Global norms for iodine level of salt in the household is 15 ppm, based on the assumption that average salt consumption of 10 g per day would provide the adult iodine requirement of 150 μg per day [1]. Salt with iodine levels of less than 5.0 ppm is considered to be non-iodised [19].

For the UIC data, the recommended WHO/UNICEF/ICCIDD [1, 19] criteria were used to characterise the iodine status of the non-pregnant women that participated in this study. According to these criteria, a population of non-pregnant women is considered iodine deficient if the median UIC is below 100.0 μg/L, and iodine sufficient if the median is in the range of 100–200 μg/L. In addition, in an iodine sufficient population, the UIC in not more than 20% of the urine samples should be below 50.0 μg/L. The median UIC can also be used to indicate the severity of iodine deficiency; for example, a population with median UIC <20.0 μg/L is considered severely deficient, and moderately or mildly deficient, if it is 20.0 to 49.0 μg/L or 50–99 μg/L, respectively [1]. It is important to note that, due to significant variation of urinary iodine levels throughout the day, it is not possible to use single urine samples to assess iodine status of an individual. Therefore, the median of urinary iodine concentrations from single

urine samples from a group or population is presented and used to indicate the iodine status of that group or population.

## Ethical approval

Ethical approval was obtained from the PNG National Department of Health Medical Research Advisory Committee (NDoH MRAC) and the Ethics and Research Grant committee in School of Medicine and Health Sciences (SMHS), University of Papua New Guinea (UPNG). Since the majority of the community cannot read or write, we obtained informed verbal consent from village authorities. Informed verbal consent was also obtained from heads of the households and from each participating woman. Participant consent was documented on the interview form. The ethics committees approved this consent procedure.

# Results

## Availability of salt in households

Of the 62 households visited, only 61 (98.4%) had salt. Assuming salt was indeed not available in the households of the women who said they did not have salt, and assuming salt availability in the 62 visited households was reflective of salt availability in the households of the women who said they did have salt, the salt availability in this community was 51.6% (98.4% x 149/ 284).

## Iodine content in salt from households

The mean (± STD) iodine content in salt samples from the households was 37.8 ± 11.8 ppm (mg/kg), 95% Confidence Interval (95% CI) was 34.8–41.8 ppm, the range was 3.6–75.5 ppm, median was 36.1 ppm and Interquartile Range (IQR) was 33.4–40.0 ppm. The iodine content was below 5.0 ppm in 1.6% (1/61) of the salt samples; i.e., <2% of salt was non-iodised. Iodine content was inadequate (5.0–29.9 ppm) in 11.5% (7/61) of the salt samples, adequate (30.0–50.0 ppm) in 75.4% (46/61) and excess (> 50.0 ppm) in 11.5% (7/61) of salt samples. Thus, 86.9% (53/61) of salt samples from the households could be considered adequately iodised (>30.0 pm), based on PNG legislation-required iodine levels at production and import and estimated 30% loss.

Using the salt iodine content criteria recommended by the WHO/IGN/UNICEF [1, 19] of greater than 15.0 ppm, 96.7% (59/61) of the households that had salt, had adequately iodised salt. However, the global strategy of universal salt iodisation (USI) assumes that all households have salt, and a target of greater than 90% of households with adequately iodised salt (> 15 ppm) has been suggested as an indicator of a successful program [1]. Assuming that the salt in the other households with salt was similarly iodised, 50.4% of all households (148 x 96.7% = 143, and 143/284 = 50.4%) in the study community had adequately iodised salt. This is far below the 90.0% recommended coverage for all households with adequately iodised salt that should indicate effective implementation of the USI strategy [1, 19].

## Iodine content in salt from markets

The mean iodine content in the salt purchased from the markets was: Kotidanga (Lot 1): 39.9 ± 0.18 ppm, Kanabea (Lot 2): 39.4 ±0.80 ppm and Ipayu (Lot 3): 39.4 ± 0.42 ppm. Thus, the combined mean iodine content in the salt purchased from the three markets was 39.6 ± 0.52 ppm, the 95% CI was 39.2–40.0 ppm, range was 38.7 to 40.3 ppm, median was 39.8 ppm and IQR was 39.3–39.9 ppm. For the salt purchased in the National Capital District, Port Moresby (NCD), which was the same brand as in the Kotidanga Rural LLG markets, the

mean iodine content was 39.7 ± 0.47 ppm, the 95% CI was 39.3–40.1 ppm, range was 39.0–40.5 ppm, median was 39.8 ppm and IQR was 39.4–39.9 ppm. No difference was observed in the mean iodine content in the salt from the Kotidanga Rural LLG markets and the NCD or the households, suggesting little loss of iodine occurs between the NCD and Kotidanga Rural LLG, or between the markets and the households.

## Discretionary per capita intake of salt and estimated per capita intake of iodine

The mean discretionary intake of salt per capita per day was 3.9 ± 1.21 g, with a range of 2.2 to 6.5 g and median of 3.4 g. As noted above, the mean iodine content in the salt collected from the households was 37.8 ppm. Thus, the calculated mean discretionary intake of iodine per capita per day was 147.4 μg, with a range of 83.2 to 245.7 μg. Assuming that 20% of the iodine in salt is lost during storage in the household and food preparation [1], the calculated per capita discretionary intake of iodine becomes 117.9 μg per day, with a range of 66.6 to 196.6 μg per day. Thus, the calculated mean discretionary daily per capita intake of iodine was within the 90.0 μg to 120.0 μg recommended daily requirement of iodine for children, but below the 150 to 200 μg recommended for non-pregnant, pregnant and lactating women [1, 2, 4].

## Socio-demographic characteristics of the non-pregnant women

The socio-demographic characteristics of the women are presented in Table 1. Over 55.0% (157/284) of the women were below 30.0 years of age. Among the 284 women only one (0.4%) graduated from university and five (1.8%) completed secondary school. None of the women worked for money.

## Urinary iodine concentration (UIC)

The Shapiro-Wilks test showed that the frequency distribution curve of the UIC (μg/L) for all the women was not normally distributed (p = 0.001). The summary statistics of the UIC for the 284 women are presented in Table 2. The median UIC was 34.0 μg/L and the 95% Confidence Interval obtained from bootstrapping was 30.0 to 38.0 μg/L. This indicates moderate status of iodine deficiency among the women at the time of this study. In addition, 66.5% (189/284) had UIC below 50.0 μg/L.

## Comparison of the UIC of non-pregnant women in households with salt and without salt

For further analysis, the UIC data was separated according to the response of the women to the question *"Does your family have salt today in the house?"* Summary statistics of the UIC data for both groups are also presented in Table 3. Both groups of women had moderate status of iodine deficiency. However, for the 148 (52.1%) women with salt in the house the median UIC was higher (39.5 μg/L) compared to the median UIC for the 136 (47.9%) with no salt in the house (29.0 μg/L), although this difference was not statistically significant (p = 0.08, 2-tailed).

The responses to the questionnaire are presented in Tables 3 and 4.

All the 284 non-pregnant women interviewed in the market responded to the questions in the questionnaire. The results show that the majority of women (95.4%) reported using something to give food a salty taste. When using a product to give food a salty taste, salt was the most commonly used product (97.4%), but Maggie/bouillon cubes and traditional salt/ash were also common, used by 39.8% and 24.0% of women, respectively. Frequency of usage was

**Table 1. Socio-demographic characteristics of non-pregnant women.**

| Parameters | Characteristics non-pregnant women |
|---|---|
| N | 284 |
| Mean age (years) | 27.4 |
| Standard deviation (STD) | 8.2 |
| 95% Confidence Interval (95% CI) | 26.4–28.4 |
| Age range (years) | 15.0–45.0 |
| Median age (years) | 25.5 |
| Interquartile range (IQR) | 20.0–35.0 |
| Age groups (years) | |
| 15–19.9 | 21.1% (60/284) |
| 20–29.9 | 34.1% (97/284) |
| 30–39.9 | 34.9% (99/284) |
| 40–44.9 | 9.9% (28/284) |
| Weight (kg) | |
| Mean | 41.0 |
| STD | 4.5 |
| 95% CI | 40.5–41.5 |
| Range | 31.0–75.0 |
| Median | 41.0 |
| IQR | 39.0–43.0 |
| Level of education | |
| University graduate | 0.4% (1/284) |
| Completed secondary school | 1.8% (5/284) |
| Completed primary school | 25.3% (72/284) |
| No formal education | 72.5% (206/284) |
| Marital status | |
| Single | 18.3% (52/284) |
| Married | 79.6% (226/284) |
| Separated/divorced/widow | 2.1% (6/284) |
| Mean number of people in household | 6.67 ± 2.5 people |

lower, however, particularly for Maggie/bouillon cubes; 59.6% used them at least once a week compared to salt, which was used at least once a week by 95% of households that use it. Maggie/bouillon cubes appear to be used as a complement to salt rather than instead of it; 38.7% of women used both Maggie/bouillon and salt, whereas only 1.5% used Maggie/bouillon only. Other results are shown in Table 3.

When asked what they would do if salt was cheaper, majority of women (71.8%) indicated they would buy salt more often or in larger quantities. However, over a quarter (26.4%)

**Table 2. Summary statistics of the urinary iodine concentration (μg/L) for all women, and for those with salt in the house and no salt in the house.**

| Parameters | All women | Salt in the house | No salt in the house |
|---|---|---|---|
| Number (%) | 284 | 148 (52.1%) | 136 (47.9%) |
| Median UIC (μg/L) | 34.0 | 39.5 | 29.0 |
| Interquartile Range (IQR) (μg/L) | 20.0–61.8 | 20.0–67.0 | 19.0–53.8 |
| 95% Confidence interval (Bootstrapping) (μg/L) | 30.0–38.0 | 32.0–47.0 | 28.0–32.0 |
| % (n) with UIC below 50 μg/L | 66.5% (189) | 61.7% (91) | 72.1% (98) |

**Table 3. Responses on use of salty flavourings and commercial salt.**

| | Questions | Responses |
|---|---|---|
| Q1 | Does your household use anything to give food a salty taste? (n = 284) | % (n) |
| | (1) Yes | 95.4 (271) |
| | (2) No | 4.6 (13) |
| | (3) Not sure | 0 |
| Q2 | If yes, what do you use? Select as many as apply (n = 271) (some participants selected more than one option) | |
| | (1) Salt | 97.4 (264) |
| | (2) Maggie/Bouillon/other cubes<br>  a. Maggie/bouillon/other cubes plus Salt<br>  b. Maggie/bouillon/other cubes only | 39.8 (109)<br>38.7 (105)<br>1.5 (4) |
| | (3) Ash/traditional salt | 24.0 (65) |
| | (4) Any other from market/shop-specify | 0 |
| Q3 | How often do you use the product (other than salt) mentioned in question 2? | |
| | Maggie/Bouillon/other cubes (n = 109) | |
| | (1) Everyday | 3.7 (4) |
| | (2) Several times a week | 6.4 (7) |
| | (3) Once a week | 49.5 (54) |
| | (4) Once a month or less | 40.4 (44) |
| | Ash/traditional salt (n = 65) | |
| | (1) Everyday | 6.2 (4) |
| | (2) Several times a week | 35.4 (23) |
| | (3) Once a week | 47.7 (31) |
| | (4) Once a month or less | 10.8 (7) |
| Q4 | Does your family have salt in the household today? (n = 284) | |
| | (1) Yes | 52.5 (149) |
| | (2) No | 47.5 (135) |
| Q5 | If No, did your household have salt yesterday? (n = 135) | |
| | (1) Yes | 3.0(4) |
| | (2) No | 97.0 (131) |
| Q6 | If No, did your household have salt any day in the last 7 days? (n = 135) | |
| | (1) Yes | 26.7(36) |
| | (2) No | 73.3 (99) |
| Q7 | How frequently do you add salt to your family's cooking? (n = 264) | |
| | (1) Every day | 25.8 (68) |
| | (2) Several times a week | 29.9 (79) |
| | (3) Once a week | 39.4 (104) |
| | (4) Once a month or less | 5.0 (13) |
| Q8 | What do you do with the salt? (n = 264) | |
| | (1) Use for cooking and add to food before eating | 35.6 (94) |
| | (2) Use for cooking only | 52.3 (138) |
| | (3) Add to food before eating only | 11.4 (30) |
| | (4) Other uses-specify | 0.8 (2; in pig food) |
| Q9 | How often do you buy salt? (n = 264) | |
| | (1) Every day | 5.7 (15) |
| | (2) Sometimes | 91.3 (241) |
| | (3) Not at all | 3.0 (8) |
| Q10 | Why do you buy salt only sometimes or not at all? (n = 249) | |

*(Continued)*

**Table 3.** (Continued)

| | Questions | Responses |
|---|---|---|
| | (1) Too expensive | 71.5 (178) |
| | (2) Not always available | 26.5 (66) |
| | (3) Do not like it | 1.6 (4) |
| | (4) Prefer to use other product to make food salty–specify | 0.4 (1; ash) |
| Q11 | If you buy salt everyday or sometimes, how do you usually buy it? (n = 256) | |
| | (1) In the small packet with a name and logo that it was originally packed in | 60.9 (156) |
| | (2) In a small bag that it was re-packed into | 24.2 (62) |
| | (3) In a small amount wrapped in newspaper or a leaf (by the shopkeeper) | 14.1 (36) |
| | (4) Loose–not packed or wrapped in anything | 0.8 (2) |
| Q12 | If salt was cheaper, what would you do? (n = 284) | |
| | (1) Buy it more often and/or buy more of it | 71.8 (204) |
| | (2) No difference | 26.4 (75) (salt not available; have asthma; do not like taste) |
| | (3) Not sure | 1.8 (5) |
| Q13 | If you ever have salt in your home, how do you usually store it? (n = 284) | |
| | (1) Never have salt at home | 5.6 (16) |
| | (2) In the bag that I bought it in | 42.3 (120) |
| | (3) In a container with a lid | 19.4 (55) |
| | (4) In open container, no lid | 3.9(11) |
| | (5) Other—specify | 28.9 (82; bamboo) |

indicated this would make no difference, as salt may not be available for purchase close to home. When asked about how they store salt at home, almost two thirds (61.7%) stored it in a way that would be expected to retain the iodine; 42.3% said they stored salt in the original bag and 19.4% stored it in a container with a lid. However, nearly a third (28.9%) stored it in a piece of bamboo, which may cause loss of iodine over time.

The women were also asked about their use of other industrially processed foods, to see if any of these might be suitable vehicles for iodine fortification, see Table 4. The results revealed, however, that the consumption of wheat flour/products made with wheat flour, oil and rice was significantly lower, than that of salt. Of the three, oil was the most commonly used; 25.7% of the women had oil for cooking in their household on the interview day, compared to only 3.5% for wheat flour/wheat flour products and 1.1% for rice, respectively. Overall use of these other processed foods was very low.

## Discussion

Global experience has demonstrated that fortification of salt with iodine is an equitable, effective and sustainable strategy to ensure optimal iodine nutrition for all population groups [19]. Accordingly, salt iodisation is the recommended long-term public health measure for prevention and control of IDD worldwide. According to WHO/UNICEF/IGN, household coverage of adequately iodised salt is one of three parameters that can be used for assessing and monitoring the implementation of salt fortification [1, 19]. Therefore, regular assays of iodine content in household salt and monitoring of the availability of adequately iodised salt in all households are important activities in national salt iodisation programs.

**Table 4. Responses on use of industrially processed foods.**

| Q14 | Does your household have wheat flour or a food made from wheat flour such as noodles, bread, crackers, biscuits, scones, and donuts today? (n = 284) | |
|---|---|---|
| | (1) Yes | 3.5 (10) |
| | (2) No | 96.5 (274) |
| Q15 | If No, did your household have wheat flour or a food made from wheat flour such as noodles, bread, crackers, biscuits, scones, or donuts yesterday? (n = 274) | |
| | (1) Yes | 1.8 (5) |
| | (2) No | 98.2 (269) |
| Q16 | If No, did your household have wheat flour any day last week or a food made from wheat flour such as noodles, bread, crackers, biscuits, scones, and donuts? (n = 274) | |
| | (1) Yes | 5.5 (15) |
| | (2) No | 94.5 (259) |
| Q17 | If you responded yes to question 14, 15 or 16, which food did you have in your household (tick any that apply)? (n = 32) | |
| | (1) Wheat flour | 3.1 (1) |
| | (2) Noodles/pasta | 81.3 (26) |
| | (3) Bread/buns/rolls | 0 |
| | (4) Crackers/biscuits | 3.1 (1) |
| | (5) Scones/donuts | 0 |
| | (6) Noodles/pasta and crackers/biscuits | 12.5 (4) |
| Q18 | Does your household have oil for cooking today? (n = 284) | |
| | (1) Yes | 25.7 (73) |
| | (2) No | 74.3 (211) |
| Q19 | If No, did your household have oil for cooking yesterday? (n = 211) | |
| | (1) Yes | 1.4(3) |
| | (2) No | 98.6 (208) |
| Q20 | If No, did your household have oil for cooking any day last week? (n = 211) | |
| | (1) Yes | 31.8 (67) |
| | (2) No | 68.2 (144) |
| Q21 | Does your household have rice today? (n = 284) | |
| | (1) Yes | 1.1 (3) |
| | (2) No | 98.9 (281) |
| Q22 | If No, did your household have rice yesterday? (n = 281) | |
| | (1) Yes | 1.1 (3) |
| | (2) No | 98.9 (278) |
| Q23 | If No, did your household have rice any day last week? (n = 281) | |
| | (1) Yes | 9.6 (27) |
| | (2) No | 90.4 (254) |

A basic premise of the WHO/UNICEF/IGN recommendation for salt iodisation as the primary strategy to increase iodine intake is that majority of households should have access to commercial salt [1]. Contrary to this basic premise, in the present study commercial salt was not available in 47.9% (136/284) of the households on the interview day. Of those who responded "no", only 3% had salt in the household the day before the interview, and 26.7% had salt in the last seven days. Reasons for not buying salt were cost (71.5%) or limited availability (26.5%). The 47.9% households with no salt in this study was higher than the 35.6% households with no salt reported in an earlier study in the same community [13], and the 38.0% of households reported nationally for the PNG National Nutrition Survey of 2005 [9]. Limited availability and affordability of salt in households appears to be a major constraint to

achieving optimal iodine intake through salt iodisation in some communities in PNG and the fact that 38% of households did not have salt on the day of data collection of the National Nutrition Survey of 2005 [9] suggests that this phenomenon may occur in a substantial proportion of communities in PNG.

Based on the PNG salt iodisation legislation requirement of a minimum of 50 ppm at production /import level (and hence, an estimated 30 ppm at household level), in this study adequately iodised salt was available in 86.9% of the randomly selected households **with salt** on the day of the visit. Compared to other studies, the 86.9% was higher than the 45.2% reported in an earlier study in this area in 2018 [13], 78.0% in Morobe district in 2013 [23] and 66% in Karimui-Nomane district in 2017 [20]; but lower than the 95.0% in Hella district in 2004 [24], 94.5% in National Capital district in 2006 [25] and 95.0% in National Capital district Port Moresby in 2009 [11]. In general, the data from this study, as well as earlier studies, suggests that when salt is available, it is mostly adequately iodised. However, as noted, because 48.4% of households did not have salt on the day of the household visit, only 45.0% (147 x 86.9% /284) of households in this community had salt adequately iodised, according to PNG legislation. According to the WHO/UNICEF/IGN criteria of at least 15 ppm iodine in salt in **all** households with salt, only 50.4% of all households had adequately iodised salt. Regardless of the iodine requirement used, the proportion of households with adequately iodised salt in this community was far below the 90.0% recommended coverage that indicates effective implementation of the salt iodisation strategy [1, 19].

The single brand of salt sold in the markets was adequately iodised, based on the PNG Food Sanitation Regulations criteria for commercial table salt [7, 8]. The mean iodine ($37.8 \pm 11.8$ ppm) content in the salt samples from the households was similar to the mean iodine ($39.6 \pm 0.52$ ppm) content in the salt from the markets. This seems to indicate minimal loss of iodine in the salt samples between the markets and the households.

The daily per capita discretionary intake of salt ($3.9 \pm 1.21$ g) was lower than the assumed 10.0 g per capita per day salt intake that formed the basis for required iodine levels specified in the PNG salt legislation [7, 8]. It is also lower than the 6.23 g reported in the NCD [9], the 6.6 g in Lae City [26], 4.7 g in Morobe [23] and 5.0 g in Simbu province [20]. It is, however, higher than the $2.9 \pm 1.8$ g per capita per day reported in a previous study in the same community [13] and 2.62 g in Hella district in 2004 [24]. It is also well below the lowest intake indicated in a 2010 assessment of sodium intakes worldwide [27].

In the households with salt, the calculated mean per capita discretionary intake of iodine was 117.9 µg. Assuming that iodised salt is added directly to the prepared food, this is lower than the 150 µg to 200 µg recommended daily requirement of iodine for non-pregnant women [1, 19]. Only a quarter (25.8%) add salt to family food every day. Further iodine loss may occur through the practice of adding iodised salt to water for cooking and then draining it, along with much of the salt (52.3%). These findings are similar to the results of the previous study [13]. Also compared to the previous study, there was no improvement in salt storage practices; in this study, only a fifth (19.4%) stored salt in a container with a lid [13]. Inadequate storage practices reduce iodine content in iodised salt, resulting in a lower iodine intake compared to the recommended daily requirement for non-pregnant women [1, 4, 19]. A very important finding is that 26.7% of households had no salt on the interview day or the previous 7 days. A significant proportion of the population in this community is thus largely unreached by salt iodisation, which is the only intervention currently being implemented in PNG to ensure optimal iodine intake. The suboptimal intake of dietary iodine by the non-pregnant women at the time of this study is concerning, because of the association between iodine deficiency and the potential risk of irreversible damage to the foetus and neonate, when these women become pregnant [1, 2, 4]. The study did find that 38.7% of households consumed Maggie/bouillon

cubes in addition to salt; 59.6% at least once per week. As Maggie/bouillon cubes are 50–70% salt, their consumption would contribute to salt intake and, provided this salt was iodised, to iodine intake. Maggie/bouillon cubes used in Ghana and other West African countries contain iodine; they have been found to contribute significantly to salt and iodine intake in these countries [28].

The median UIC of 34.0 μg/L indicates moderate iodine deficiency among the study participants. In addition, the UIC was below 50.0 μg/L in 66.5% of the women. Moderate iodine deficiency was also reported among non-pregnant women in an earlier study, carried out in the same community in 2015 [12]. The median UIC was 36.0 μg/L, and 57.7% of the women had UIC below 50.0 μg/L. Although the median UIC in both studies are similar, the percent of women (66.5%) with UIC below 50.0 μg/L is higher in the present study, compared to the result (57.7%) obtained in the same community in a mini survey conducted in 2015 [12]. The median UIC of school age children in the same community was found to be 32 μg/L in 2015 and 64.2% of the children had UIC below 50.0 μg/L [12]. Follow-up study conducted in the same community in 2017 found the median UIC of school age children to be 25.5 μg/L and 76.5% of the children had UIC below 50.0 μg/L [13]. These data together confirm persistent iodine deficiency in the general population of Kotidanga Rural LLG; iodine status in pregnant and lactating women is likely to be worse because of their higher iodine requirements.

The median UIC among the women in households with salt (39.5 μg/L) was higher than that of women in households with no salt (29.0 μg/L). The difference was, however, not statistically significant. The suboptimal status of iodine nutrition among the non-pregnant and non-lactating women, despite the availability of adequately iodised salt in the markets, is likely due to low availability of commercial salt in households (47.9% had no salt on the interview day) and low salt consumption, even when salt was provided for free (to the sub-sample of the households in which discretionary salt intake was assessed).

In the present study, results of respondents show that women have low educational attainment, mainly practicing subsistence farming. Low education level and remoteness may contribute to the apparent lack of awareness of the need to consume adequate amounts of iodine for optimal growth and development [1, 4]. A recent review of access to iodised salt in ten low resource countries found that access to iodised salt was inequitable in all participating countries. Low socio economic status and rural residence were found to be risk factors for low household access to salt [29].

To achieve optimal iodine nutrition among the non-pregnant and non-lactating women in the study area, increased intake of dietary iodine is paramount. The major requirement to increase dietary iodine in PNG is to improve salt availability and affordability. In addition, raising public awareness and education of the health benefits of iodised salt, including improvement in salt storage and meal preparation practices may be useful. Some of these strategies have been used to address IDD in remote and urban communities in other resource-limited countries, including Ethiopia [30], Pakistan [31, 32] and India [33, 34].

The assessment of availability of processed foods for potential fortification with iodine indicated that consumption of wheat-based products and rice is low. This may be due to the high cost and limited availability of these products. Usage of Maggie/bouillon was reported by a considerable proportion of women (39.8%); however, of those that used it, only 59.6% used it at least once per week, indicating that Maggie/bouillon usage is also low, lower even than that of commercial salt, which was available in 62.9% of households in the week prior. Moreover, the majority of Maggie/bouillon consumers used it in addition to salt. It can be assumed that a combination of household salt and Maggie/bouillon, made with iodised salt, could contribute significantly to both salt and iodine intake in this community. At least a third of households in the study population appear to be benefiting from this consumption pattern. Consumption of

edible vegetable oil was moderate (25.7% had oil on the interview day). The findings confirm that salt is most likely the optimal food vehicle for fortification with iodine in this remote community.

As stated in an earlier study at the present site [13], the limited access to commercial salt is a major limiting factor in the effectiveness of salt iodisation in this community. Subsidising transport costs, improving distribution networks, and making salt available in smaller, more affordable quantities could potentially increase availability and household usage of salt. It is important that adequate iodisation of most of the salt, which was demonstrated by this study, is maintained. In addition, distribution of iodine or micronutrient supplements to women of reproductive age in remote communities would improve iodine intake if adequate and sustained distribution could be achieved and monitored [35].

## Conclusion

In this remote community, there is insufficient consumption of adequately iodised salt, as a consequence of limited access to commercial salt, mainly due to remoteness, cost and availability. The study found that the commercial salt that was available was adequately iodised as per national requirements and global recommendations. Thus, the limited access to, and the low consumption of commercial salt appear to be the reasons for moderate iodine deficiency among non-pregnant women in this remote community. Moreover, other mini-surveys suggest this constraint to the salt iodisation strategy is replicated in other remote communities in PNG. This is a significant public health concern that requires multi-strategy interventions, in particular, increasing access to and availability of commercial salt and iodised Maggie/bouillon cubes in remote communities, continuing to ensure that the salt is adequately iodised, reviewing the required salt iodine levels in consideration of salt intake patterns, and ensuring the use of iodised salt in the production of Maggie/bouillon cubes. Commitment at all levels of government is essential for the successful implementation of such strategies.

## Acknowledgments

The authors thank the Kotidanga Rural LLG village authorities, women and health staff for their participation in this study. We acknowledge the support of the Chief Technical Officer and other technical staff members in the Division of Basic Medical Sciences, School of Medicine and Health Sciences, University of University of Papua New Guinea.

The findings and conclusions in this manuscript are those of the authors; they do not represent the official position of the institutions and authors' organisations. The authors declare no conflicts of interest. A small research grant was received from the Iodine Global Network for the fieldwork section of this project. The analyses of iodine in salt and urine were funded by and carried out in the Micronutrient Research Laboratory, School of Medicine and Health Sciences, University of Papua New Guinea.

## Author Contributions

**Conceptualization:** Janny M. Goris, Victor J. Temple.

**Data curation:** Janny M. Goris.

**Formal analysis:** Janny M. Goris, Victor J. Temple, Joan Sumbis, Nienke Zomerdijk, Karen Codling.

**Investigation:** Janny M. Goris, Victor J. Temple, Joan Sumbis.

**Methodology:** Janny M. Goris, Victor J. Temple, Joan Sumbis.

**Project administration:** Janny M. Goris.

**Supervision:** Janny M. Goris, Victor J. Temple.

**Validation:** Janny M. Goris, Nienke Zomerdijk, Karen Codling.

**Visualization:** Janny M. Goris.

**Writing – original draft:** Janny M. Goris, Victor J. Temple, Joan Sumbis, Nienke Zomerdijk, Karen Codling.

**Writing – review & editing:** Janny M. Goris, Victor J. Temple, Joan Sumbis, Nienke Zomerdijk, Karen Codling.

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
