## [Decision Letter · Decision Letter 0]

3 Jul 2019

PONE-D-19-14979

Iodine nutrition status among non-pregnant women, awareness of iodine and available food vehicles for fortification with iodine in Kotidanga Rural LLG, Kerema district, Gulf province, Papua New Guinea

PLOS ONE

Dear Dr Temple,

Thank you for submitting your manuscript to PLOS ONE. After careful consideration, we feel that it has merit but does not fully meet PLOS ONE’s publication criteria as it currently stands. Therefore, we invite you to submit a revised version of the manuscript that addresses the points raised during the review process.

All the three reviewers recommended important revision of this manuscript, including one rejection. The authors have a chance to improve a lot the manuscript addressing all comments and suggestions.

We would appreciate receiving your revised manuscript by Aug 17 2019 11:59PM. To enhance the reproducibility of your results, we recommend that if applicable you deposit your laboratory protocols in protocols.io, where a protocol can be assigned its own identifier (DOI) such that it can be cited independently in the future. For instructions see: http://journals.plos.org/plosone/s/submission-guidelines#loc-laboratory-protocols

We look forward to receiving your revised manuscript.

Kind regards,

Marly A. Cardoso, Ph.D.

Academic Editor

PLOS ONE

Journal Requirements:

2. Please include a copy of the questionnaire used in the study, in both the original language and English, as Supporting Information, or include a citation if it has been published previously.

Reviewers' comments:

Reviewer's Responses to Questions

**Comments to the Author**

1. Is the manuscript technically sound, and do the data support the conclusions?

Reviewer #1: Partly

Reviewer #2: Partly

Reviewer #3: No

2. Has the statistical analysis been performed appropriately and rigorously? 

Reviewer #1: Yes

Reviewer #2: No

Reviewer #3: Yes

3. Have the authors made all data underlying the findings in their manuscript fully available?

Reviewer #1: Yes

Reviewer #2: Yes

Reviewer #3: Yes

4. Is the manuscript presented in an intelligible fashion and written in standard English?

Reviewer #1: Yes

Reviewer #2: Yes

Reviewer #3: No

5. Review Comments to the Author

Reviewer #1: This paper describes the iodine status, use of iodised salt ad level of salt iodisation in women of child bearing age in PNG. this would be of interest to those working in the field.

Title

The title is overly long and does not include iodised salt intake. Need to make the location information more brief.

Also use the term iodine status not iodine nutrition status.

Introduction

Line 52-55 - If you are discussing iodine status in pregnancy and lactation surely all these could affected in the fetus and the infant. Make this clear.

Line 75-79 - In the 2005 PNG National nutrition survey, how is it possible for 92.5% to have iodised salt on day of collection and yet 38% had no salt. This needs clarification. The UIC in this study is adequate (median 170 mcg) yet this was some time ago, what is the UIC for the whole population now?

Line 82-83 - You state UIC was low in NDC after the National Nutrition Survey. Is this correct, this contradicts lines 92-94 when you state iodine intake is adequate for most of the population, but not in remote communities. I assume NDC is not a remote community.

For the other surveys discussed in PNG can you provide the median UIC.

Line 98-99 You need to make it clearer why this research adds to your previous research. Your study aim needs to be clearer, and not contain details which belong in the methods.

Methods

Line 166-167 – were children and adults in the house treated equally? If children eat less of the food they will have less exposure and adults more? Is this a validated method to measure salt exposure?

Line 171 – what time of day were the samples collected? We know that there is diurnal variation in UIC and it is lower in the morning.

Line 214 – what is the difference between table salt and other salt? What are these other salts? Perhaps participants were using other salt which has a lower permitted range?

Results

There is substantial repetition within the results section. Do not to include information already provided in the methods.

The information about the person who reported having salt and then did not have any is repeated numerous times throughout the methods and results. Write this once, then classify 136 as households without salt and 148 households with salt.

Line 248 – remove “because of logistical reasons…” this is repetition.

Line 266-270 – remove you have stated this previously

Line 300-309 – this data would be better in a table. When reporting medians always give Q1, Q3.

Table 1 – This is misleading. Surely the person who reported having salt and then who had non needs to be reclassified as no salt in house and the data recalculated accordingly.

Table 2 is very long, you need to pick out what is important and put in a clear table

Discussion

There is some repetition in this section also you state twice that 33% of households had no salt in the house in the last week (line 378 and 425)

Line 406 – this is not correct. Although the means were similar the range and SD were very different, so you cannot make this assumption. Because of the huge range in the salt from the homes, do it not seem more likely they are from different sources and not just this sample in the market?

Line 414-415 – why do you think the use is so low, you need to comment on this. Could it be the use of bouillon as salt? Or do you feel you method inadequately measured salt intake?

Line 433 are bouillon iodised in other countries? Make this clear.

Line 453 - it is not clear what you mean by “even when salt was provided freely”?

Reviewer #2: SUMMARY

Using the survey data collected in a remote area in Gulf Province, Papua New Guinea, this study assessed the iodine status of non-pregnant women, awareness and use of iodized salt, and the availability of other industrially processed foods that maybe fortified with iodine. This study found there was insufficient consumption of adequately iodized salt in this remote area, mainly resulting from low access to and limited consumption of commercial salt due to remoteness, cost, and availability.

REVIEWS COMMENTS:

1. Sampling

In lines 127-132, the authors stated that

“According to the recently released UNICEF Guidance on the Monitoring of Salt Iodization Programmes and Determination of Population Iodine Status [19]“ around 400 urine samples per population group are required to measure the median UIC with 5% precision and 100 urine samples to measure the median UIC with 10% precision”. In the current mini-survey with limited resources, a sample size of 300 non-pregnant women of childbearing age was considered adequate to provide sufficient precision to determine the median UIC.”

The study surveyed 300 women aged 15 to 45 years old who visited the major markets. Out of 300, 16 observations, who were pregnant women in their first trimester, were excluded from the analysis.

My concerns are as follows:

1) Were the women who visited the major markets systematically different from those who did not visited the markets? If so, how could the authors generalize their results?

2) Reporting error was partly addressed in the study. For example, a subsample (N=62) was randomly selected to check for idolized salt at home among a total of 149 households who reported to have such salt at home. However, there may be a case -- households who reported not to have idolized salt at home actually had such salt. Therefore, the estimates

2. Methods and Analyses

The authors conducted the Shapiro-Wilks test for normality. The result shows that the frequency distribution of the UIC (ug/L) for all women was not normally distributed (p-value = 0.0001) (see Line 313). However, it was not clear whether the authors assume normal distribution for some results in Table 1 as well as in the subsection titled “Comparison of the UIC of non-pregnant women in households with salt and without salt.” If the normal distribution was not assumed, what was assume for the distribution for the statistical analyses?

Reviewer #3: The study addresses an important nutritional risk of iodine nutrition status for women of child-bearing age in a rural area of Papua New Guinea, and provides support for developing strategies to improve dietary intake of iodine through fortification. My major concerns relate to the lack of evidence on “awareness of iodine” and on measurement of per capita dietary intake of iodine.

Comments:

1. Line 76-79. The household reference is not clearly stated. Line 76 refers to 92.5% of households having salt. Line 79 indicates 38% of households had no salt. Please clarify. Also, which groups of households are being compared to have the lower/higher iodine status?

2. Line 161. Please reword the statement on the selection of households. The 20 households were not “…randomly selected…and visited for collection of a teaspoon of salt.” Perhaps better would be: “…among the 62 households visited for collection of a teaspoon of salt.”

3. Lines 161, lines 293-298 and line 409 and following. These sections report on the method of obtaining data on average discretionary intake of salt per capita. No demographic information on the household other than count of individuals was obtained. The estimate of average per capita intake makes no adjustment of children in that count. More young children in the household would mean that the “average per capita intake” for the woman would likely be underestimated. Some note of caution in comparison to the national standards should be considered. I note that the ranges are given for children and for women in the description of results (lines 295-290).

4. Lines 177 and following. Please state how many questionnaires were completed. Based on the statement in lines 340-341, this seems to be a questionnaire administered to all of the women sampled in the market (n=284).

5. Lines 262 and following. Information in this paragraph is quite repetitive. As example: lines 262-263 and lines 271-272. Also, the calculation presented in line 273 should be more carefully stated. Better would be: (148 x 96.7% = 143 and 143/284 = 50.4%).

6. Lines 332 and Table 2. The title of the manuscript and this section indicate there are results on “salt iodine awareness”. However, there is very little information to elicit information on “awareness” of iodine. On looking at the supporting information for the questionnaire (S3), I note Q12 reads “If iodized salt was cheaper, what would you do?”. This follows Q10 which asks why do you buy salt only sometimes or not at all, with the first option being “too expensive”. Without any further information, it seems unlikely that the “awareness” is specifically about iodized salt. I suggest dropping the reference to “awareness” in title and text unless there is more information provided to respondents than noted. Also, as listed now in Table 2, Q12 -- the word “iodized” is missing from the question.

7. Lines 458-459. Related to the previous comment, based on the single question about the use of iodized salt (as presented similar to other questions about salt being expensive), the statement that “low education level and remoteness” as contributing to lack of awareness about consuming adequate amounts of iodine goes significantly beyond the data collected for this study.

Minor edits:

Line 355: Please edit: “…in a piece of bamboo from which there is may be loss…”

Line 456: Suggest “A majority of women…” or “The majority of women…”

6. PLOS authors have the option to publish the peer review history of their article (what does this mean?). If published, this will include your full peer review and any attached files.

Reviewer #1: No

Reviewer #2: No

Reviewer #3: No

---

## [Author Response · Author response to Decision Letter 0]

12 Aug 2019

. Review Comments to the Author

Reviewer #1: This paper describes the iodine status, use of iodised salt ad level of salt iodisation in women of child bearing age in PNG. This would be of interest to those working in the field.

Title

The title is overly long and does not include iodised salt intake. Need to make the location information more brief.

Also use the term iodine status not iodine nutrition status.

Response:

We have modified the title of the manuscript as suggested by the two reviewers. However we did not include “iodised salt intake” so as to reduce the length of the title. The new title is:

“Iodine status of non-pregnant women and availability of food vehicles for fortification with iodine in a remote community in Gulf province, Papua New Guinea”

Introduction

Line 52-55 - If you are discussing iodine status in pregnancy and lactation surely all these could affected in the fetus and the infant. Make this clear.

Response:

We have amended the sentence and added “and infant”: “During pregnancy and lactation, adequate intake and bioavailability of iodine are required for the biosynthesis of thyroid hormones, which are important for the regulation of growth and the healthy development of the nervous system of the foetus and infant, control of metabolic activities, developmental processes, and functions of the central nervous system.”

Line 75-79 - In the 2005 PNG National nutrition survey, how is it possible for 92.5% to have iodised salt on day of collection and yet 38% had no salt. This needs clarification. 

Response:

We agree with the reviewer, the sentence is incorrect and misleading. The sentence has been corrected. The paragraph should read therefore: “Findings in the National Nutrition Survey in PNG in 2005 (PNG NNS, 2005) indicated that adequately iodised salt was available in 92.5% of households with salt. It further stated that normal status of iodine nutrition was prevalent among non-pregnant women of childbearing age with Median Urinary Iodine Concentration (Median UIC) of 170µg/L [9]. However, on the day of data collection, 38% of households had no salt, and women in those households had lower iodine status than those in households with salt (median UIC of 114µg/L and 203µg/L respectively)”

The UIC in this study is adequate (median 170 mcg) yet this was some time ago, what is the UIC for the whole population now?

Response:

No other national survey has been carried out since the 2005 National nutrition survey.

Line 82-83 - You state UIC was low in NDC after the National Nutrition Survey. Is this correct, this contradicts lines 92-94 when you state iodine intake is adequate for most of the population, but not in remote communities. 

Response:

Yes, the information is correct. We are referring to the results of the two mini-surveys carried out in the National Capital District after the National Nutrition Survey; the references for those mini-surveys [10, 11] are indicated in the reference section. This information does not contradict the information in lines 92-94 which is referring to the current data available from PNG. We have rephrased the sentence thus:

“Considering currently available PNG data on iodine status and availability of iodised salt, it appears that, while the majority of household salt in the country is adequately iodised and iodine status of the general population is adequate, there are remote communities that have very limited access to commercial and, therefore, iodised salt, and that these remote communities, as a consequence, suffer from mild to moderate iodine deficiency”.

I assume NDC is not a remote community.

Response:

Yes, your assumption is correct. The National Capital District (NCD), not NDC, is the commercial center of PNG. Port Moresby, which is the capital of PNG, is located in the NCD. 

For the other surveys discussed in PNG can you provide the median UIC.

Response: 

Yes, the Median UIC obtained for some mini surveys in PNG are presented in ANNEX 1 (pages 12-13 below). 

Line 98-99 You need to make it clearer why this research adds to your previous research. Your study aim needs to be clearer, and not contain details which belong in the methods.

Response:

We have amended the section by stating more clearly the specific objectives of the study and deleted details about the methods used:

“The objectives of the current study were to assess the iodine status of non-pregnant women, availability and use of commercial salt, extent to which it is iodised, and availability of other industrially processed foods suitable for fortification with iodine”.

Methods

Line 166-167 – were children and adults in the house treated equally? If children eat less of the food they will have less exposure and adults more? Is this a validated method to measure salt exposure?

Response:

We agree with the issues raised by the reviewer. The adults were counted not children. The word “individuals” used in the text was an error; “individuals” has been replaced by “adults”, because only adults were counted. The 24-hour urinary sodium excretion, which is the best method for assessing the per capita intake of sodium (salt), was not used because of logistical reasons. 

Yes, the method we used is a validated method. It is one of the recommended indirect methods: “Weighed household salt/salt disappearance studies, where a household’s salt container (or salt provided by the study) is weighed at the start and after a specific period of time (e.g. seven days) and the difference in salt is divided by the number of days and household members”

[See Reference: WHO Regional Office. Using dietary intake modelling to achieve population salt reduction: A guide to developing a country-specific salt reduction model, WHO 2018. Available from

http://www.euro.who.int/__data/assets/pdf_file/0004/365242/salt-report-eng.pdf?ua=1]

Line 171 – what time of day were the samples collected? We know that there is diurnal variation in UIC and it is lower in the morning.

Response:

The standard protocol for assessing UIC in a population does not put any restriction on the time of urine collection; the usual practice is collection of single urine sample from each individual selected randomly. The UIC is not a parameter for the diagnosis of iodine status in an individual; it is an epidemiologic parameter for assessing iodine status in a community. 

However, in our present study, urine samples were collated between 11.00 am and 4.00 pm. [See Reference: WHO, UNICEF, ICCIDD. Assessment of Iodine Deficiency Disorders and monitoring their elimination: A guide for programme managers. Geneva: WHO/NHD; 2007. Available from: http://www.who.int/nutrition/publications/micronutrients/iodine_deficiency/9789241595827/en/]

Line 214 – what is the difference between table salt and other salt? What are these other salts? Perhaps participants were using other salt which has a lower permitted range?

Response:

Table salt is the refined free-flowing salt that contains about 95 to 99% of Sodium Chloride. It is usually used for cooking and/or added to ready-to-eat food. In PNG all edible salt for human and animal use, including table salt, course/crystal salt, should be fortified with potassium iodate. Sea salt, which contains little natural iodine, can be processed and used as table salt. 

In PNG some communities use traditional salt made locally (inorganic ash from incineration of parts of some plants which has no iodine-this salt was tested by us in a previous study). In the questionnaire the answer to Q 2 indicated that they use “ash / traditional salt”, which does not contain iodine. 

Results

There is substantial repetition within the results section. Do not to include information already provided in the methods.

Response: 

We agree with the reviewer. We have gone through the text and made all the changes and modifications recommended. We have also tried to reduce the repetition to a minimum. 

The information about the person who reported having salt and then did not have any is repeated numerous times throughout the methods and results. Write this once, then classify 136 as households without salt and 148 households with salt.

Response:

The suggested changes have been made in the text and table. 

Line 248 – remove “because of logistical reasons…” this is repetition.

Response: 

We agree with the reviewer. The phrase was deleted and the sentence modified to read thus: “Therefore, the study selected a sub-sample of 62 households (41.6%) from the 149. These 62 households were visited for collection of a teaspoon of salt”.

Line 266-270 – remove you have stated this previously

Response:

We agree with the reviewer. We have deleted the four sentences. 

Line 300-309 – this data would be better in a table. When reporting medians always give Q1, Q3.

Response:

As requested we have put the Socio-demographic characteristics in in a table (see Table 1). We have also included the Interquartile Range (IQR) with the Median. 

Table 1 – This is misleading. Surely the person who reported having salt and then who had non needs to be reclassified as no salt in house and the data recalculated accordingly.

Response:

We accepted the suggestion by the reviewer. Accordingly we have recalculated the results and made all the corrections in Table 2. Please note that the values for the 95% CI (Bootstrapping) did not change. 

Table 2 is very long, you need to pick out what is important and put in a clear table

Response:

We agree with the reviewer. However, we do not wish to delete any of the questions and their answers because, in our view, all the questions are relevant to the problems affecting availability and use of iodised salt in remote communities. In addition, one of the reviewers requested that we provide the full questionnaire and data. 

Discussion

There is some repetition in this section also you state twice that 33% of households had no salt in the house in the last week (line 378 and 425).

Response:

Line 378 has been deleted. The 33% has been corrected to 26.7% as indicated in the answer to question Q6. 

Line 406 – this is not correct. Although the means were similar the range and SD were very different, so you cannot make this assumption. Because of the huge range in the salt from the homes, do it not seem more likely they are from different sources and not just this sample in the market?

Response:

We appreciate the observation and opinion of the reviewer, but we wish to maintain our assumption based on the results obtained. In the results section we have made corrections to the iodine content in salt samples from the households and the markets. We have also included the interquartile ranges (IQR) to the median of the samples. For the salt from the households the median iodine content was 36.1 ppm and IQR was 33.4 to 40.0 ppm. For the salt from the markets the median iodine content was 39.8 ppm and the IQR was 39.3 to 39.9 ppm. 

The huge range in the iodine content in salt samples from the homes compared to the range in salt samples from the markets or shops is a common phenomenon that has been reported by various researchers. One of the major reasons is because of the different ways of storage of iodised salt in the households as reported in our present findings (Table 3, Q13). In the markets and shops the salts are kept in their original packages, thus reducing the chances of loss of iodine. 

Line 414-415 – why do you think the use is so low, you need to comment on this. 

Response: 

The low consumption may be due to low availability of commercial salt and high cost and, dislike for the taste of salt. Due to cultural reasons and tradition little salt is added to traditional food during preparation. 

Could it be the use of bouillon as salt?

Response:

Usage of Maggie/bouillon was reported by a considerable proportion of women (39.8%), however, of those that used it, only 59.6% used it at least once per week, indicating that Maggie/bouillon usage is also low; lower even than commercial salt, which was available in 62.9% of households in the week prior to the time of the interview. Nevertheless, when used, the majority of Maggie/bouillon users used it in addition to salt.

Or do you feel you method inadequately measured salt intake?

Response:

One of the standard recommended procedures was used to measure the discretionary intake of salt. 

[See Reference: WHO Regional Office. Using dietary intake modelling to achieve population salt reduction: A guide to developing a country-specific salt reduction model, WHO 2018. Available from

http://www.euro.who.int/__data/assets/pdf_file/0004/365242/salt-report-eng.pdf?ua=1]

Line 433 are bouillon iodised in other countries? Make this clear.

Response: 

Yes, in other countries bouillon and Maggie cubes are made with iodised salt, thus they contain iodine. The sentence has been modified accordingly: “Maggie/bouillon cubes used in Ghana and other West African countries contain iodine; they have been found to contribute significantly to salt and iodine intake in these countries [28]”

Line 453 - it is not clear what you mean by “even when salt was provided freely”?

Response:

This is referring to the method section of this manuscript under sub-heading “Discretionary intake of salt”. Lines 173 to 176 reads: “A sealed 100 g packet of the same brand of salt found in the market was given to each of the 20 randomly selected households to use for food preparation and consumption as usual. The number of adults living in each household and eating food from the same cooking pot/hearth was counted and recorded. Each household was visited three days later to determine the amount of salt remaining in the packet”.

Reviewer #2: SUMMARY:

Using the survey data collected in a remote area in Gulf Province, Papua New Guinea, this study assessed the iodine status of non-pregnant women, awareness and use of iodized salt, and the availability of other industrially processed foods that maybe fortified with iodine. This study found there was insufficient consumption of adequately iodized salt in this remote area, mainly resulting from low access to and limited consumption of commercial salt due to remoteness, cost, and availability.

REVIEWS COMMENTS:

1. Sampling

In lines 127-132, the authors stated that

“According to the recently released UNICEF Guidance on the Monitoring of Salt Iodization Programmes and Determination of Population Iodine Status [19]“ around 400 urine samples per population group are required to measure the median UIC with 5% precision and 100 urine samples to measure the median UIC with 10% precision”. In the current mini-survey with limited resources, a sample size of 300 non-pregnant women of childbearing age was considered adequate to provide sufficient precision to determine the median UIC.”

The study surveyed 300 women aged 15 to 45 years old who visited the major markets. Out of 300, 16 observations, who were pregnant women in their first trimester, were excluded from the analysis.

My concerns are as follows:

1) Were the women who visited the major markets systematically different from those who did not visited the markets? If so, how could the authors generalize their results?

Response:

Indeed, we agree with the reviewer that this issue is very important. However, this is a study carried out in a remote mountainous area with limited road access, thus the decision to focus on the markets as the best option to get representative sampling of the women in the community. This is a community where most people visit markets to exchange or trade excess agricultural produces and for social contacts. However, it is possible some women never or hardly ever visit markets, and some women who visit much less frequently than others, and consequently have less chance of being included in the study. Women who don’t visit markets or who visit less frequently are likely to have even lower usage of commercial salt and an even higher risk of iodine deficiency. Such women may be more remote, more traditional, less educated and less well-off 

2) Reporting error was partly addressed in the study. For example, a subsample (N=62) was randomly selected to check for idolized salt at home among a total of 149 households who reported to have such salt at home. However, there may be a case -- households who reported not to have idolized salt at home actually had such salt. Therefore, the estimates

Response: 

Indeed, it is possible that households who reported not to have iodised salt at home actually had salt. However this is very unlikely as salt is an expensive commodity for this community. In addition, this is an inaccessible mountainous area without infrastructure. Most of the houses are located at great distances from each other. Coming to the markets involves walking long distances along mountain tracks, often with landslides and crossing creeks without bridges. Thus, we think that our estimate is quite reasonable and representative of the situation in this remote community. 

2. Methods and Analyses

The authors conducted the Shapiro-Wilks test for normality. The result shows that the frequency distribution of the UIC (ug/L) for all women was not normally distributed (p-value = 0.0001) (see Line 313). However, it was not clear whether the authors assume normal distribution for some results in Table 1 as well as in the subsection titled “Comparison of the UIC of non-pregnant women in households with salt and without salt.” If the normal distribution was not assumed, what was assumed for the distribution for the statistical analyses?

Response:

“Table 2: Summary statistics of the urinary iodine concentration (µg/L) for all the women, and for those with salt in the house and no salt in the house”

The UIC data for all the women and for the two groups of women (those in households with salt and without salt) were tested using the Shapiro-Wilks test for normality. The data for each of the groups was not normally distributed. Thus, Non-parametric statistics were used for analyses of the three sets of data. The median and IQR values were calculated as appropriate. The 95% confidence interval (Bootstrapping) is the additional parameter recommended in the new UNICEF guidelines for presentation of UIC results (See Reference: “UNICEF. Guidance on the Monitoring of Salt Iodization Programmes and Determination of Population Iodine Status; New York: UNICEF; 2018. Available from: http://www.unicef.org/nutrition/files/Monitoring-of-Salt-Iodization.pdf ”.)

Reviewer #3: 

The study addresses an important nutritional risk of iodine nutrition status for women of child-bearing age in a rural area of Papua New Guinea, and provides support for developing strategies to improve dietary intake of iodine through fortification. My major concerns relate to the lack of evidence on “awareness of iodine” and on measurement of per capita dietary intake of iodine.

Comments:

1. Line 76-79. The household reference is not clearly stated. Line 76 refers to 92.5% of households having salt. Line 79 indicates 38% of households had no salt. Please clarify. 

Response:

We agree with reviewer #3; the sentence is incorrect and misleading. The sentence has been corrected. The paragraph should read thus: “Findings in the National Nutrition Survey in PNG in 2005 (PNG NNS, 2005) indicated that 92.5% of salt samples taken from households were adequately iodised. It further stated that iodine status was “adequate” among non-pregnant women of childbearing age with Median Urinary Iodine Concentration (UIC) of 170 µg/L [9]. However, on the day of data collection, 38% of households had no salt, and women in those households had lower iodine status than those in households with salt (median UIC of 114µg/L and 203 µg/L respectively)”

Also, which groups of households are being compared to have the lower/higher iodine status?

Response: 

The two groups are, non-pregnant women in households with salt (Median UIC of 170 �g/L and non-pregnant women in households with no salt (Median UIC of 114�g/L).

2. Line 161. Please reword the statement on the selection of households. The 20 households were not “…randomly selected…and visited for collection of a teaspoon of salt.” Perhaps better would be: “…among the 62 households visited for collection of a teaspoon of salt.”

Response:

We have rephrased the sentence as suggested. The sentence reads therefore: “A subset of 20 households was randomly selected among the 62 households and visited for collection of a teaspoon of salt”.

3. Lines 161, lines 293-298 and line 409 and following. These sections report on the method of obtaining data on average discretionary intake of salt per capita. No demographic information on the household other than count of individuals was obtained. The estimate of average per capita intake makes no adjustment of children in that count. More young children in the household would mean that the “average per capita intake” for the woman would likely be underestimated. Some note of caution in comparison to the national standards should be considered. I note that the ranges are given for children and for women in the description of results (lines 295-290).

Response:

We agree with the issues raised by the reviewer. The adults were counted not children. The word “individuals” used in the text was an error; “individuals” has been replaced by “adults”, because only adults were counted. 

The 24-hour urinary sodium excretion, which is the best method for assessing the per capita intake of sodium (salt), was not used because of logistical reasons. 

Yes, we used one of the indirect methods recommended: “Weighed household salt/salt disappearance studies, where a household’s salt container (or salt provided by the study) is weighed at the start and after a specific period of time (e.g. seven days) and the difference in salt is divided by the number of days and household members”

[See Reference: WHO Regional Office. Using dietary intake modelling to achieve population salt reduction: A guide to developing a country-specific salt reduction model; WHO 2018. Available from

http://www.euro.who.int/__data/assets/pdf_file/0004/365242/salt-report-eng.pdf?ua=1]

4. Lines 177 and following. Please state how many questionnaires were completed. 

Response: 

All the 284 questionnaires were completed. This information is already in the text as indicated in your sentence below.

Based on the statement in lines 340-341, this seems to be a questionnaire administered to all of the women sampled in the market (n=284).

Response:

Yes, this sentence is already in the text “All the 284 non-pregnant women interviewed in the markets responded to the questions in the questionnaire”. This gave a response rate of 100%. 

5. Lines 262 and following. Information in this paragraph is quite repetitive. As example: lines 262-263 and lines 271-272.

Response:

We agree with the reviewer, thus we have deleted the sentence: “Of the 61 households visited which did have salt, 96.7% of the salt samples had greater than 15 ppm of iodine and were considered adequately iodised”

Also, the calculation presented in line 273 should be more carefully stated. Better would be: (148 x 96.7% = 143 and 143/284 = 50.4%).

Response:

We agree with the reviewer. We have made the change as suggested. 

6. Lines 332 and Table 2. The title of the manuscript and this section indicate there are results on “salt iodine awareness”. However, there is very little information to elicit information on “awareness” of iodine. On looking at the supporting information for the questionnaire (S3), I note Q12 reads “If iodized salt was cheaper, what would you do?”. This follows Q10 which asks why do you buy salt only sometimes or not at all, with the first option being “too expensive”. Without any further information, it seems unlikely that the “awareness” is specifically about iodized salt. 

Response: 

We agree with the reviewer. Q4 to Q12 are questions about practices in use of salt. We have changed the title of the manuscript, the heading and description of table 4 (renamed table 3).

I suggest dropping the reference to “awareness” in title and text unless there is more information provided to respondents than noted. 

Response:

We have removed “awareness” from the title of the manuscript, and also the heading and description of table 3. 

(Table 3. Responses on use of salty flavourings, commercial salt and industrially processed foods)

Also, as listed now in Table 2, Q12 -- the word “iodized” is missing from the question.

Response:

We agree with the reviewer. However, the word “iodized” was not used frequently in the questionnaire because an earlier study in this geographical area showed that only iodised commercial salt was sold in the markets. 

7. Lines 458-459. Related to the previous comment, based on the single question about the use of iodized salt (as presented similar to other questions about salt being expensive), the statement that “low education level and remoteness” as contributing to lack of awareness about consuming adequate amounts of iodine goes significantly beyond the data collected for this study.

Response:

Low education level and remoteness are major contributing factors to lack of awareness. 

The data indicating low education level is presented under the subtitle “Socio-demographic characteristics of the non-pregnant women”. As requested by one of the reviewers the data is presented in Table 1 (now Table 2). It shows that 72.5% of the women had no formal education, 1.8% completed secondary school and only 0.4% has a university degree. 

With regards to the remoteness: The actual study site is a remote mountainous area with very limited/no roads to get there. The way of getting there is mainly by walking along mountain paths or by air (helicopter) transport, which is very expensive. It usually takes about 3 days walking during daylight from the location of the hospital to the closest settlement in Kerema, Gulf province or Menyamya in Morobe province.

Minor edits:

Line 355: Please edit: “…in a piece of bamboo from which there is may be loss…”

Response:

The sentence has been edited. 

Line 456: Suggest “A majority of women…” or “The majority of women…”

Response:

The sentence has been edited. 

6. PLOS authors have the option to publish the peer review history of their article (what does this mean?). If published, this will include your full peer review and any attached files.

Response:

We have no objection to publishing the peer review history of our article. 

Do you want your identity to be public for this peer review? For information about this choice, including consent withdrawal, please see our Privacy Policy.

Reviewer #1: No

Reviewer #2: No

Reviewer #3: No

ANNEX 1: Information requested by Reviewer I:

MEDIAN URINARY IODINE CONCENTRATION (UIC) FOR SOME MINI SURVEYS IN PNG

1. Ref: Amoa B, Rubiang L, Iodine status of pregnant women in Lae. Asia Pac J Clin Nutr 2000; 9 (1): 33 – 35.

• Lae city Morobe Province PNG

o Pregnant women Median UIC: 231.0�g/L

2. Ref: Temple VJ, Mapira P, Adeniyi KO, Sims P. Iodine deficiency in Papua New Guinea (Sub-clinical iodine deficiency and salt iodization in highlands of Papua New Guinea). J of Public Health, 2005, 27: 45 – 48.

• Tari and Koroba districts Hella Region Southern Highlands Province PNG

o School age children Median UIC: 48.0�g/L

o Male children Median UIC: 67.0�g/L

o Female children Median UIC: 44.0�g/L

3. Ref: Department of Health of Papua New Guinea, Unicef Papua New Guinea, University of Papua New Guinea, US Centres of Disease Control and Prevention. Papua New Guinea National Nutrition Survey 2005; Pac J Med Sci. 2011; 8(2): 54-9.

• National Nutrition Survey, PNG

o Among non-pregnant women (15-49 years) Median UIC: 183.75�g/L,

o In households without salt on the day of the survey Median UIC: 122.2�g/L

o In households in clusters without any salt Median UIC: 91.7�g/L, 

4. Ref: Temple VJ, Haindapa B, Turare R, Masta A, Amoa AB and Ripa P. Status of Iodine Nutrition in Pregnant and Lactating Women in National Capital District, Papua New Guinea. Asia Pacific Journal of Clinical Nutrition, 2006; 15 (4): 533 – 537.

• National Capital District (NCD) PNG:

o Non-pregnant women Median UIC: 163.0�g/L

o Lactating women Median UIC: 134.0�g/L

o Pregnant women Median UIC: 180.0�g/L

• Pregnant women:

o First Trimester Median UIC: 165.0�g/L

o Second Trimester Median UIC: 221.5�g/L

o Third Trimester Median UIC: 178.0�g/L

5. Ref: Temple VJ, Oge R, Daphne I, Vince JD, Ripa P, Delange F and Eastman CJ. “Salt Iodization and Iodine Status among Infants and Lactating Mothers in Papua New Guinea” AJFAND, Vol. 9, No. 9, Dec 2009, 1807 – 1823

• National Capital District (NCD) PNG:

o Non-pregnant women Median UIC: 169.5�g/L

o Lactating mothers Median UIC: 124.5�g/L

o All Infants Median UIC: 253.5�g/L

o Exclusively breast-fed infants Median UIC: 251.0�g/L

o Mothers of Exclusively breast-fed infants Median UIC: 117.5�g/L

o Mixed-fed infants Median UIC: 290.0�g/L

o Mothers of Mixed-fed infants Median UIC: 155.0�g/L

6. Ref: Lomutopa SJ, Aquame C, Willie N and VJ Temple; Status of Iodine Nutrition among School-age Children (6 – 12 years) in Morobe and Eastern Highlands Provinces, Papua New Guinea, Pacific J. Medical Sciences 2013, Vol. 11, No. 2, 70 – 87.

• Aseki-Menyamya district Morobe province and Gouno, Mt. Michael Local-Level Government area in Lufa district Eastern Highlands province, PNG.

• Aseki-Menyamya district, 

o School age children Median UIC: 149.5�g/L

o Male children Median UIC: 145.8�g/L

o Female children Median UIC: 168.0�g/L

• Gouno Lufa district Mt. Michael Local-Level Government area {remote community} 

o School age children Median UIC: 50.0μg/L 

o Male children Median UIC: 51.3�g/L

o Female children Median UIC: 46.3�g/L

7. Ref: Goris J, Zomerdijk N, Temple V. Nutritional status and dietary diversity of Kamea in Gulf province, Papua New Guinea. Asia Pac J Clin Nutr. 2017; 26(4):665-70. doi: 10.6133/apjcn.052016.09.

• Kamea Gulf Province {Remote community}

o School age children Median UIC: 32.0�g/L

o Non-pregnant women Median UIC: 36.0�g/L

8. Ref: Goris J, Temple V, Zomerdijk N, Codling K. Iodine status of children and knowledge, attitude, practice of iodised salt use in a remote community in Kerema district, Gulf province, Papua New Guinea. PLoS ONE. 2018; 13(11):e0197647. doi: https://doi.org/10.1371/journal.pone.0197647.

• Kamea Gulf Province {Remote community}

o School age children: Median UIC: 25.5�g/L

9. Ref: Temple V, Kiagi G, Kai H, Namusoke H, Codling K, Dawa L, et al. Status of iodine nutrition among school-age children in Karimui-Nomane and Sina-Sina Yonggomugl districts in Simbu province, Papua New Guinea. Pac J Med Sci. 2018; 18(1):3-20.

• Simbu Province 

• Karimui-Nomane {Remote community}:

o School age children: Median UIC: 17.5μg/L 

o Male children: 16.5μg/L 

o Female children: 15.5μg/L

• Sina Sina Yonggomugl {Remote community}

o School age children: Median UIC: 57.5μg/L

o Male children: Median UIC: 61.3μg/L

o Female children: Median UIC: 53.5μg/L.

---

## [Decision Letter · Decision Letter 1]

9 Sep 2019

PONE-D-19-14979R1

Iodine status of non-pregnant women and availability of food vehicles for fortification with iodine in a remote community in Gulf province, Papua New Guinea

PLOS ONE

Dear Dr Temple,

Thank you for submitting your manuscript to PLOS ONE. After careful consideration, we feel that it has merit but does not fully meet PLOS ONE’s publication criteria as it currently stands. Therefore, we invite you to submit a revised version of the manuscript that addresses the points raised during the review process.

Please consider the revision based on the remaining reviewer's comments. 

We would appreciate receiving your revised manuscript by Oct 24 2019 11:59PM. To enhance the reproducibility of your results, we recommend that if applicable you deposit your laboratory protocols in protocols.io, where a protocol can be assigned its own identifier (DOI) such that it can be cited independently in the future. For instructions see: http://journals.plos.org/plosone/s/submission-guidelines#loc-laboratory-protocols

We look forward to receiving your revised manuscript.

Kind regards,

Marly A. Cardoso, Ph.D.

Academic Editor

PLOS ONE

Reviewers' comments:

Reviewer's Responses to Questions

**Comments to the Author**

1. If the authors have adequately addressed your comments raised in a previous round of review and you feel that this manuscript is now acceptable for publication, you may indicate that here to bypass the “Comments to the Author” section, enter your conflict of interest statement in the “Confidential to Editor” section, and submit your "Accept" recommendation.

Reviewer #1: All comments have been addressed

Reviewer #3: (No Response)

2. Is the manuscript technically sound, and do the data support the conclusions?

Reviewer #1: Yes

Reviewer #3: Yes

3. Has the statistical analysis been performed appropriately and rigorously? 

Reviewer #1: Yes

Reviewer #3: Yes

4. Have the authors made all data underlying the findings in their manuscript fully available?

Reviewer #1: Yes

Reviewer #3: Yes

5. Is the manuscript presented in an intelligible fashion and written in standard English?

Reviewer #1: Yes

Reviewer #3: Yes

6. Review Comments to the Author

Reviewer #1: All the suggested changes have been addressed to a satisfactorily standard. I have no further comments.

Reviewer #3: The authors have addressed most of my first round comments adequately. Thank you for attention to the details of my comments and those of the other reviewers. I have a few remaining comments.

1. Lines 149-155 (and in reference to results presented in lines 276-286, and also 398-402). I note that all markets had the one brand of commercial salt that was then analysed for iodine content and comparison. Because all of the salt from a single brand was likely produced in the same facility (or at least under common company guidance), it may be useful to provide an additional sentence on the market share or importance of this brand. Although recognizing the market regulation, one might expect that more variation exists across brands than within a single brand. This concern is of less importance if this brand represents a major share of salt sold commercially in the country (or survey region).

2. Table 3. I agree with a previous comment about the length of Table 3. One possible approach to making the length more manageable would be to break the table into 2, with Q14 and following questions in a second table. The second table with focus on the use of “other processed foods” (as described line lines 351-356).

3. Lines 453-456. Related to previous comment #7. Thank you for being more careful throughout about the information on “awareness” that had been previously emphasized in title and discussion. To be more precise here, I suggest rewording line 454-455 to read: “Low education level and remoteness may contribute [i.e., not “may have contributed”] to the apparent lack of awareness…” That makes more clear that there is no finding in the analysis presented on “awareness” per se.

Suggested edits:

Line 36: change to: “Salt was available on the interview day…”

Line 67: change to: “Salt iodisation has been implemented…”

Lines 77-78: change to: “…38% of household had no salt in the household, and women in these households had lower…”

Lin 486: drop comma to read: “…would improve iodine intake if adequate and sustainable…” This is an essential condition to the statement.

7. PLOS authors have the option to publish the peer review history of their article (what does this mean?). If published, this will include your full peer review and any attached files.

Reviewer #1: No

Reviewer #3: No

---

## [Author Response · Author response to Decision Letter 1]

5 Oct 2019

Reviewers' comments:

Reviewer's Responses to Questions

Comments to the Author:

1. If the authors have adequately addressed your comments raised in a previous round of review and you feel that this manuscript is now acceptable for publication, you may indicate that here to bypass the “Comments to the Author” section, enter your conflict of interest statement in the “Confidential to Editor” section, and submit your "Accept" recommendation.

Reviewer #1: All comments have been addressed

Reviewer #3: (No Response)

Response:

Many thanks for this feedback.

2. Is the manuscript technically sound, and do the data support the conclusions?

Reviewer #1: Yes

Reviewer #3: Yes

Response:

Many thanks for this feedback.

3. Has the statistical analysis been performed appropriately and rigorously? 

Reviewer #1: Yes

Reviewer #3: Yes

Response:

Many thanks for this feedback.

4. Have the authors made all data underlying the findings in their manuscript fully available?

Reviewer #1: Yes

Reviewer #3: Yes

Response:

Many thanks for this feedback.

5. Is the manuscript presented in an intelligible fashion and written in standard English?

Reviewer #1: Yes

Reviewer #3: Yes

Response:

Many thanks for this feedback.

6. Review Comments to the Author

Reviewer #1: All the suggested changes have been addressed to a satisfactorily standard. I have no further comments.

Reviewer #3: The authors have addressed most of my first round comments adequately. Thank you for attention to the details of my comments and those of the other reviewers. I have a few remaining comments.

1. Lines 149-155 (and in reference to results presented in lines 276-286, and also 398-402). I note that all markets had the one brand of commercial salt that was then analysed for iodine content and comparison. Because all of the salt from a single brand was likely produced in the same facility (or at least under common company guidance), it may be useful to provide an additional sentence on the market share or importance of this brand. Although recognizing the market regulation, one might expect that more variation exists across brands than within a single brand. This concern is of less importance if this brand represents a major share of salt sold commercially in the country (or survey region). 

Response:

We agree with the observation of reviewer #3. However, we cannot comment on the market share of this brand of salt or the importance of this brand. We can only state that at the time of this study only one brand of salt was available in the market, unlike an earlier study in which three different brands of salt were available in the markets (see reference below). As already stated, the study site is a remote inaccessible mountainous area without infrastructure, and there is no road access. The way of getting there is mainly by walking along mountain paths or by air (helicopter) transport, which is very expensive. It usually takes about 3 days walking during daylight from the location of the hospital to the closest settlement in Kerema, Gulf province or Menyamya in Morobe province. This may imply that the various commercial items including brand(s) of salt in the markets depend on what the traders were able to purchase and at what cost and how fast they are able to make profit from the sale of their products. 

Ref: “Goris J, Temple V, Zomerdijk N, Codling K. Iodine status of children and knowledge, attitude, practice of iodised salt use in a remote community in Kerema district, Gulf province, Papua New Guinea. PLoS ONE. 2018; 13(11):e0197647. doi: https://doi.org/10.1371/journal.pone.0197647”

2. Table 3. I agree with a previous comment about the length of Table 3. One possible approach to making the length more manageable would be to break the table into 2, with Q14 and following questions in a second table. The second table with focus on the use of “other processed foods” (as described line lines 351-356).

Response:

As suggested by reviewer #3 we have divided Table 3 into two separate tables, with Table 3 covering questions 1 to 13, including responses on use of salty flavourings and commercial salt and Table 4, covering questions 14 to 23, including responses to the use of industrially processed foods.

Table 3: Responses on use of salty flavourings and, commercial salt

Table 4: Responses to use of industrially processed foods

3. Lines 453-456. Related to previous comment #7. Thank you for being more careful throughout about the information on “awareness” that had been previously emphasized in title and discussion. To be more precise here, I suggest rewording line 454-455 to read: “Low education level and remoteness may contribute [i.e., not “may have contributed”] to the apparent lack of awareness…” That makes more clear that there is no finding in the analysis presented on “awareness” per se.

Response:

We have changed the text as suggested by reviewer #3.

“Low education level and remoteness may contribute to the apparent lack of awareness of the need to consume adequate amounts of iodine for optimal growth and development ………”

Suggested edits:

Line 36: change to: “Salt was available on the interview day…”

Line 67: change to: “Salt iodisation has been implemented…”

Lines 77-78: change to: “…38% of household had no salt in the household, and women in these households had lower…”

Lin 486: drop comma to read: “…would improve iodine intake if adequate and sustainable…” This is an essential condition to the statement.

Response:

We have made all the edits in the text (line 36, line 67, lines 77-78 and line 486) as suggested by reviewer #3. 

7. PLOS authors have the option to publish the peer review history of their article (what does this mean?). If published, this will include your full peer review and any attached files.

Do you want your identity to be public for this peer review? For information about this choice, including consent withdrawal, please see our Privacy Policy.

Reviewer #1: No

Reviewer #3: No

---

## [Editor Report · Decision Letter 2]

9 Oct 2019

Iodine status of non-pregnant women and availability of food vehicles for fortification with iodine in a remote community in Gulf province, Papua New Guinea

PONE-D-19-14979R2

Dear Dr. Temple,

We are pleased to inform you that your manuscript has been judged scientifically suitable for publication and will be formally accepted for publication once it complies with all outstanding technical requirements.

With kind regards,

Marly A. Cardoso, Ph.D.

Academic Editor

PLOS ONE
---

## [Editor Report · Acceptance letter]

7 Nov 2019

PONE-D-19-14979R2 

Iodine status of non-pregnant women and availability of food vehicles for fortification with iodine in a remote community in Gulf province, Papua New Guinea 

Dear Dr. Temple:

I am pleased to inform you that your manuscript has been deemed suitable for publication in PLOS ONE. Congratulations! Your manuscript is now with our production department. 

With kind regards,

on behalf of

Dr. Marly A. Cardoso 

Academic Editor

PLOS ONE